Genome-wide characterization and expression of the TLP gene family associated with Colletotrichum gloeosporioides inoculation in Fragaria × ananassa

Zhang Yuchao
Miao Lixiang
Yang Xiaofang
Jiang Guihua jgh2004267@sina.com
The Institute of Horticulture, ZheJiang Academy of Agricultural Sciences , Hangzhou, Zhejiang , China
Adhikari Tika
Electronic publication date: 2022 Mar 24
Publication date: 2022
Volume: 10
Electronic Location ID: e12979
Received 2021 Sep 1; Accepted 2022 Jan 31
Copyright: © 2022 Zhang et al.
Copyright year: 2022
Copyright holder: Zhang et al.
License: This is an open access article distributed under the terms of the Creative Commons Attribution License, which permits unrestricted use, distribution, reproduction and adaptation in any medium and for any purpose provided that it is properly attributed. For attribution, the original author(s), title, publication source (PeerJ) and either DOI or URL of the article must be cited.
License URL: https://creativecommons.org/licenses/by/4.0/

Keywords: Fragaria × ananassa, Thaumatin like-proteins, Colletotrichum gloeosporioides, Gene family, Expression analysis, Antifungal activity

Funding: Science and Technology Program of Zhejiang Province LY20C150003, 2021C02066-7, 2021XTTGSC02-2, 2021SNLF009 This work was supported by Science and Technology Program of Zhejiang Province (No. LY20C150003, No. 2021C02066-7, No. 2021XTTGSC02-2, No. 2021SNLF009). The funders had no role in study design, data collection and analysis, decision to publish, or preparation of the manuscript.

==============================
Background

Colletotrichum gloeosporioides, a soil-borne fungal pathogen, causes significant yield losses in many plants, including cultivated strawberry (Fragaria × ananassa, 2n = 8x = 56). Thaumatin-like proteins (TLPs) are a large and complex family of proteins that play a vital role in plant host defense and other physiological processes.

Methods

To enhance our understanding of the antifungal activity of F. × ananassa TLPs (FaTLP), we investigated the genome-wide identification of FaTLP gene families and their expression patterns in F. × ananassa plants upon pathogen infection. Moreover, we used RNA sequencing (RNA-seq) to detect the differences in the expression patterns of TLP genes between different resistant strawberry cultivars in response to C. gloeosporioides infection.

Results

In total, 76 TLP genes were identified from the octoploid cultivated strawberry genome with a mean length of 1,439 bp. They were distributed on 24 F. × ananassa chromosomes. The FaTLP family was then divided into ten groups (Group I–X) according to the comparative phylogenetic results. Group VIII contained the highest number of TLP family genes. qRT-PCR analysis results indicated that FaTLP40, FaTLP41, FaTLP43, FaTLP68, and FaTLP75 were upregulated following C. gloeosporioides infection in the resistant octoploid strawberry.

Conclusions

The data showed some differences in TLP gene expression patterns across different resistant strawberry cultivars, as well as faster TLP defense responses to pathogenic fungi in resistant cultivars. This study will aid in the characterization of TLP gene family members found in octoploid strawberries and their potential biological functions in plants’ defenses against pathogenic fungi.

Introduction

Thaumatin-like proteins (TLPs) are important members of the highly complex gene family pathogenesis-related protein group 5 (PR5). They are highly homologous with sweet-tasting thaumatin protein produced by Thaumatococcus daniellii fruit (Wel & Loeve, 1972; Velazhahan, Datta & Muthukrishnan, 1999; Christensen et al., 2002; Loon, Rep & Pieterse, 2006). TLPs are abundant and highly diversified in plants (Loon, Rep & Pieterse, 2006; O’Leary, Poulis & von Aderkas, 2007; Liu, Zamani & Ekramoddoullah, 2010). According to a survey conducted in the UniProt database (https://www.uniprot.org/uniprot/?query=Thaumatin-like+proteins&sort=score), 1,816 TLPs (1,720 angiosperms, 82 gymnosperms, seven bryophytes, and seven algae species) from 187 different species have been identified (Faillace et al., 2019; Jesús-Pires et al., 2020). A great number of isoforms activated by biotic or abiotic stresses and multiple rounds of distinct gene duplication events were observed across the TLP gene family’s expansion and evolution in plants, suggesting that this superfamily has a complex pattern of molecular evolution (Liu, Zamani & Ekramoddoullah, 2010; Cao et al., 2016). Previous studies have indicated that plant TLPs consist of approximately 200 amino acids, have a molecular weight of 21–26 kD (Velazhahan, Datta & Muthukrishnan, 1999), and contain 10 to 16 conserved motifs (Hulo et al., 2008). These proteins exhibit a series of responses to biotic and abiotic stress factors in plants, such as pathogen invasion, drought, wounding, freezing, and salinity (Petre et al., 2011). They also play roles in a variety of physiological and developmental processes, including organ formation, fruit ripening, and seed germination (Salzman et al., 1998; Seo et al., 2008).

The octoploid cultivated strawberry (Fragaria × ananassa, 2n = 8x = 56) is an economically important perennial horticultural crop (FAO, IFAD, UNICEF, WFP, WHO, 2017) that is widely cultivated in China, but the main cultivar is ‘Benihoppe’ from Japan. ‘Benihoppe’ is susceptible to a range of diseases including anthracnose, which is one of the most destructive fungal diseases that results in considerable losses in strawberry production (Hammerschlag et al., 2006; Dean et al., 2012), especially at the seedling stage and early stages following transplanting. On the east coast of China, anthracnose in strawberry is mainly caused by the fungus Colletotrichum gloeosporioides (Zhang et al., 2017), which can infect all aerial plant parts, and the most severe symptoms are dwarf-stem and foliar lesions.

Pathogenesis-related (PR) proteins, including TLPs, exhibit significant antifungal activity in plants (Velazhahan, Datta & Muthukrishnan, 1999). TLP gene expression is responsive to infection by a variety of fungal pathogens, such as Colletotrichum, Podosphaera, Phytophthora, Fusarium, and Neurospora (Woloshuk et al., 1991; Narasimhan et al., 2003; Liu, Zamani & Ekramoddoullah, 2010; Rather et al., 2015). TLPs play multiple roles in inhibiting hyphal growth and spore germination of various pathogenic fungi (Roberts & Selitrennikoff, 1990; Woloshuk et al., 1991; De Freitas et al., 2011), binding β-1,3-glucans to destroy the stability of fungal membranes (Grenier, Potvin & Asselin, 2000; Osmond et al., 2001; Zareie, Melanson & Murphy, 2002), and stimulating plant defensive responses against pathogenic fungi to prevent fungal damage or reduce disease symptoms. Because of their effective antifungal activity, TLP genes are promising candidates for improving plant disease resistance. Overexpression of defense-related TLP genes in transgenic Arabidopsis (Rout, Nanda & Joshi, 2016), rice (Datta et al., 1999), tobacco (Singh et al., 2013), grape (He et al., 2017), wheat (Mackintosh et al., 2007), and potato (Acharya et al., 2013) significantly enhanced plant resistance to fungal diseases.

Resistance to a variety of pathogens has been a key breeding purpose in the development of commercial strawberry cultivars. Research on plant defensive responses against fungal pathogens indicate that TLPs are promising candidates for developing better anthracnose resistance in strawberry. Although the TLP gene has been studied in a variety of plants, the composition of the TLP gene family members in octoploid strawberry and their potential defense response to the invasion of pathogenic fungi, particularly C. gloeosporioides, remain largely unexplored. Additionally, many studies on strawberry anthracnose have focused on fungicides used for disease control, but not the molecular resistance mechanisms and engineered plant disease resistance of the octoploid strawberry.

This study aimed to identify the TLP gene family members in octoploid cultivated strawberry (F. × ananassa), and examine their transcription and expression characteristics during pathogenic fungi infection. The purpose of this study was to expand our understanding of the function of strawberry TLP genes in defense responses against C. gloeosporioides.

Materials and Methods

Identification and characterization of TLPs in F. × ananassa

A hidden Markov model (Eddy, 1998) was constructed using HMMER software (version 3.0) based on the TLP sequences of Arabidopsis (https://www.arabidopsis.org) and rice (https://rapdb.dna.affrc.go.jp). It was then used as query sequences to identify homologs in F. × ananassa (https://www.rosaceae.org/organism/24345). We used BLAST (https://blast.ncbi.nlm.nih.gov/Blast.cgi) to retrieve TLP sequences for F. × ananassa with an expectation value of e−10. After filtering out the redundant sequences, the aligned sequences were considered candidate sequences for further analysis. The SMART (http://smart.embl-heidelberg.de/) and Pfam (http://xfam.org/) databases were used to verify our identification results, and only the sequence containing the TLP domain (PF00314) was considered the final FaTLP sequence (Ivica, Tobias & Peer, 2012; El-Gebali et al., 2018). We used the ExPASy ProtParam tool (http://web.expasy.org/protparam/) to predict the biophysical properties of the final FaTLP proteins (Gasteiger et al., 2003), including their relative molecular weight (kDa) and theoretical isoelectric point (pI).

Phylogenetic analysis of TLP genes

We used the amino acid sequences of the TLP proteins identified in F. × ananassa, Fragaria vesca, Arabidopsis, and rice in the phylogenetic analysis. One representative sequence was selected from each clade of Arabidopsis and rice TLPs according to the phylogeny of these two species (Shatters et al., 2006; Zhao & Su, 2010), which we then used as reference sequences in our phylogenetic analysis. The F. vesca sequences were obtained from the Genome Database for Rosaceae (GDR; https://www.rosaceae.org/species/fragaria_vesca/genome_v4.0.a1), and the Arabidopsis and Oryza sequences were obtained from The Arabidopsis Information Resource (TAIR) (http://arabidopsis.org) and The Institute for Genomic Research (TIGR) (http://rice.plantbiology.msu.edu/blast.shtml) database respectively. ClustalW (http://www.clustal.org/) (Larkin et al., 2007) was used for multiple sequence alignment. MEGA 6.0 (http://www.megasoftware.net/) (Tamura et al., 2013) was used to perform a phylogenetic analysis of the aligned protein sequences using the neighbor-joining method with 1,000 bootstrap replicates.

Protein motif analysis

The conserved motifs of the TLP proteins in F. × ananassa were predicted using MEME software (http://meme.nbcr.net/meme/, v4.11.0) and the following criteria: minimum motif width of 6, maximum motif width of 200, and maximum number of motifs set at 20.

Chromosome distribution analysis

To determine the physical locations of FaTLP genes, we established the starting positions of all FaTLP genes identified from the F. × ananassa genome database. A diagram of the chromosome locations of FaTLP genes was generated using MG2C (http://mg2c.iask.in/mg2c_v2.0/).

Plant materials

‘Benihoppe’ (susceptible) and ‘Kaorino’ (resistant) strawberry seedlings (Mangandi, Peres & Whitaker, 2015; Han et al., 2019) were cultivated at the Zhejiang Academy of Agricultural Sciences. Experimental plants of the two strawberry cultivars were propagated from runners, rooted in 10-cm diameter pots, and grown in a dedicated room with a 16 h photoperiod. The average daytime temperature was 25.0 ± 1.5 °C while the nighttime temperature was around 18.0 ± 1.5 °C with 75% relative humidity. No fungicides were applied and fertilizer was applied proportionally as needed.

C. gloeosporioides infection

The pathogenic fungus C. gloeosporioides was cultured and provided by the Institute of Plant Protection and Microbiology, Zhejiang Academy of Agricultural Sciences. For the propagation of C. gloeosporioides, we used oatmeal agar (OA) medium and the following steps: we weighed 30 g of oatmeal (Solarbio, https://www.solarbio.com/), boiled it for 1 h, and then strained it using four layers of gauze. We added 17 g of agar to the filtrate, adjusted the volume to 1 L using sterile water, autoclaved it at 121 °C for 15 min, and then divided it into petri dishes following sterilization. After cooling down, the pathogenic fungus was propagated on OA medium and stored for 7–10 days at 28 °C. The propagated spores were washed and collected with sterile water, and the concentration was adjusted to 4 × 106 spores/mL with sterile distilled water. The prepared C. gloeosporioides fungal suspension was used for inoculation. Three hundred plants were divided into three subgroups (control, susceptible, and resistant) with 100 plants each. Sixty-day-old strawberry seedlings were used for fungal inoculation. The strawberry leaves in the susceptible and resistant groups were sprayed with the spore suspension (4 × 106 spores/mL) using an atomizer until dripping. Control plants were similarly inoculated with sterile water. Twenty randomly-selected leaves from each group were sampled at 2, 6, 12, 24, and 48 h, and at 3, 4, 5, 6, and 7 d after inoculation. The samples were immediately frozen in liquid nitrogen and stored at −80 °C for further processing. Three replicates were sampled at each time point.

Transcriptome analysis

To determine the transcriptome profile of different resistant strawberry cultivars in response to C. gloeosporioides, 12 samples were used for RNA-seq analysis with three replicates per treatment: ‘Kaorino’-infected (24 h post-inoculation, hpi), ‘Kaorino’-uninfected, ‘Benihoppe’-infected (24 hpi), and ‘Benihoppe’-uninfected. The RNA-seq transcriptome library was prepared using the TruSeq RNA Sample Preparation Kit (Illumina, San Diego, CA, USA). We used HisAT2 (v2.1.0) (Kim et al., 2019) for sequence alignment and an annotated genome (Fragaria × ananassa Camarosa Genome Assembly v1.0.a1), available from the GDR (https://www.rosaceae.org/organism/24345), as a reference. The fragments per kilobase million (FPKM) value (Malone & Oliver, 2011) was used to identify differentially expressed genes (DEGs) between the two different samples. DESeq2 software (Anders & Huber, 2010) was used for differential expression analysis.

RNA isolation and quantitative RT-PCR (qRT-PCR) analysis

Total RNA was isolated using the modified CTAB method (Chang et al., 2007). The integrity of the RNA samples was examined using a U-0080D Protein nucleic acid spectrophotometer (Hitachi, Tokyo, Japan). cDNA was synthesized from 2 μg of total RNA using TranScript® II One-Step gDNA Removal and cDNA Synthesis SuperMix (TransScript®, Beijing, China). qRT-PCR was then carried out in a LightCycler® 96 real-time PCR system (Roche, Basel, Switzerland) with a DNA Green Master (Roche, Basel, Switzerland). The primers used for the validation of DEGs are shown in Table S5. The Actin gene was used as the reference gene. Each sample was repeated in triplicate.

Statistical analysis

Statistical analysis was carried out using SPSS 16.0 software (SPSS Inc., Chicago, IL, USA). The significance level was p < 0.05.

Results

Genome-wide identification of TLP genes in F. × ananassa

To understand the potential roles of TLPs in strawberry, we used cultivated strawberry (F. × ananassa) for genome-wide identification and characterization of TLP genes. We identified a total of 76 TLP gene members from F. × ananassa (designated as ‘FaTLP’), which was a greater amount than the number of TLP genes in many other plant species (De Jesus-Pires et al., 2020). Among the 76 FaTLPs identified, FaTLP68 was the longest (over 4,370 bp), FaTLP5 was the shortest (384 bp), and the mean length was approximately 1,439 bp. The molecular weights of these TLP genes ranged from 13.36 to 138.40 kD, with pI values between 4.27 and 8.77. Most of these genes were 200–400 aa in length with one to two introns and two to three exons, although several genes had over 10 introns/exons. Detailed information on the TLP genes, including their names, coding sequence (CDS) lengths, molecular weights, and pI values, can be found in Table 1.

Table 1 Physical and molecular characteristics of TLP genes in F. × ananassa.

Gene name	Scaffold location (bp)	Chromosome
position	Subgroup	Length
(bp)	Size (aa)	MW
(Da)	pI	Extron	Intron	
FaTLP1	7,737,020	7,738,607	Chr1-2	VII	1,344	310	31,952.79	4.66	3	2	
FaTLP2	24,918,873	24,919,901	Chr2-1	VI	1,029	342	35,319.38	4.41	1	0	
FaTLP3	6,717,318	6,718,064	Chr3-4	III	747	248	26,625.50	8.58	1	0	
FaTLP4	1,053,888	1,055,785	Chr5-1	V	1,218	405	43,590.76	5.43	2	1	
FaTLP5	9,55,336	9,55,816	Chr5-4	V	384	127	13,364.01	6.79	2	1	
FaTLP6	18,563,346	18,565,549	Chr6-1	VIII	1,208	332	35,850.27	8.61	2	1	
FaTLP7	33,108,008	33,109,182	Chr6-1	VIII	1,033	250	26,542.12	8.27	2	1	
FaTLP8	30,176,529	30,177,597	Chr7-1	VIII	858	285	30,512.75	7.8	3	2	
FaTLP9	31,754,141	31,755,199	Chr7-1	VIII	780	259	27,642.45	7.36	4	3	
FaTLP10	24,146,688	24,154,818	Chr7-2	V	3,289	848	92,352.63	5.06	10	9	
FaTLP11	4,809,914	4,815,227	Chr7-4	V	3,069	792	87,475.76	5.44	9	8	
FaTLP12	21,872,342	21,874,277	Chr1-1	VI	1,493	295	30,430.31	4.77	3	2	
FaTLP13	21,880,819	21,883,234	Chr1-1	VII	1,540	409	42,778.42	6.31	5	4	
FaTLP14	7,745,129	7,747,037	Chr1-2	VI	1,468	295	30,413.37	4.88	3	2	
FaTLP15	6,305,276	6,306,911	Chr1-3	VII	1,409	314	32,383.19	4.53	3	2	
FaTLP16	6,313,153	6,315,086	Chr1-3	VI	1,480	295	30,476.41	5.05	3	2	
FaTLP17	5,551,432	5,554,101	Chr1-4	VII	1,469	393	41,477.24	5.96	5	4	
FaTLP18	5,559,900	5,561,737	Chr1-4	VI	1,385	296	30,473.33	4.79	3	2	
FaTLP19	17,180,554	17,182,275	Chr2-1	X	1,219	286	30,586.86	8.05	3	2	
FaTLP20	25,893,167	25,894,288	Chr2-1	IX	1,039	228	23,591.69	4.56	2	1	
FaTLP21	3,82,116	3,84,430	Chr2-1	VII	1,714	335	34,244.90	4.29	3	2	
FaTLP22	3,89,384	3,92,027	Chr2-1	VI	1,443	276	29,395.70	8.11	3	2	
FaTLP23	24,923,609	24,925,890	Chr2-1	VI	1,596	311	32,878.54	5.33	4	3	
FaTLP24	22,298,735	22,301,183	Chr2-2	VI	1,323	253	26,818.66	7.83	2	1	
FaTLP25	22,313,583	22,315,865	Chr2-2	VII	1,694	335	34,269.89	4.29	3	2	
FaTLP26	4,205,420	4,209,653	Chr2-2	X	1,452	388	41,391.17	8.57	7	6	
FaTLP27	1,581,587	1,583,723	Chr2-3	X	1,633	286	30,503.79	8.03	3	2	
FaTLP28	22,764,585	22,767,122	Chr2-3	VI	1,401	276	29,459.78	7.83	3	2	
FaTLP29	22,784,341	22,786,568	Chr2-3	VII	1,628	340	34,734.50	4.27	3	2	
FaTLP30	2,891,948	2,893,222	Chr2-3	IX	1,187	255	26,322.57	4.29	2	1	
FaTLP31	4,012,565	4,015,252	Chr2-3	VI	2,015	372	39,197.11	4.95	3	2	
FaTLP32	25,284,514	25,286,825	Chr2-4	VI	1,707	373	39,232.11	4.96	3	2	
FaTLP33	25,287,065	25,288,842	Chr2-4	VI	819	272	28,126.24	4.28	2	1	
FaTLP34	19,300,457	19,310,292	Chr2-4	X	1,620	429	45,971.35	8.76	9	8	
FaTLP35	5,05,512	5,18,409	Chr2-4	VI	2,885	319	34,141.17	8.26	5	4	
FaTLP36	10,599,943	10,601,571	Chr5-1	I	1,484	253	27,092.00	8.77	2	1	
FaTLP37	9,829,098	9,830,623	Chr5-2	I	1,400	253	27,084.96	8.77	2	1	
FaTLP38	18,257,006	18,258,448	Chr5-3	I	1,317	253	27,088.95	8.77	2	1	
FaTLP39	24,102,586	24,103,807	Chr5-3	VIII	1,076	247	26,010.34	5.16	2	1	
FaTLP40	26,365,436	26,371,141	Chr5-3	V	2,208	735	79,611.59	6.51	3	2	
FaTLP41	9,73,897	9,76,302	Chr5-4	V	1,278	425	46,444.21	6.74	2	1	
FaTLP42	18,566,292	18,567,674	Chr6-1	VIII	1,266	237	25,411.68	5.78	2	1	
FaTLP43	24,059,925	24,061,244	Chr6-1	VIII	1,016	244	25,726.88	4.46	2	1	
FaTLP44	27,284,404	27,285,539	Chr6-1	II	1,050	244	25,699.44	7.39	2	1	
FaTLP45	3,114,373	3,115,524	Chr6-1	IV	1,055	259	28,086.23	8.26	2	1	
FaTLP46	32,540,333	32,541,588	Chr6-1	VIII	1,114	250	26,542.12	8.27	2	1	
FaTLP47	30,036,263	30,037,271	Chr6-2	IV	910	259	28,148.17	7.38	2	1	
FaTLP48	33,609,425	33,610,486	Chr6-3	II	976	244	25,706.48	7.39	2	1	
FaTLP49	2,574,069	2,574,953	Chr6-3	IV	740	244	26,497.36	7.91	3	2	
FaTLP50	63,531	64,471	Chr6-4	IV	842	262	28,599.78	8.27	2	1	
FaTLP51	11,952,207	11,953,479	Chr6-4	VIII	991	244	25,783.89	4.35	2	1	
FaTLP52	18,037,077	18,040,378	Chr6-4	VIII	1,481	247	26,227.58	4.75	2	1	
FaTLP53	9,212,045	9,213,306	Chr6-4	II	1,176	244	25,651.40	7.39	2	1	
FaTLP54	25,239,035	25,240,789	Chr7-1	II	1,053	290	30,926.20	8.03	3	2	
FaTLP55	25,879,973	25,884,142	Chr7-1	V	2,994	658	72,264.64	5.11	8	7	
FaTLP56	25,867,595	25,875,423	Chr7-1	V	2,577	858	94,842.94	5.84	13	12	
FaTLP57	23,075,906	23,077,305	Chr7-2	II	873	290	30,932.29	8.21	3	2	
FaTLP58	23,519,637	23,522,646	Chr7-2	V	2,045	651	71,453.60	5.25	7	6	
FaTLP59	24,128,411	24,131,354	Chr7-2	V	1,941	646	71,288.52	5.42	8	7	
FaTLP60	29,017,016	29,018,415	Chr7-2	II	873	290	30,886.26	8.21	3	2	
FaTLP61	1,564,471	1,566,282	Chr7-3	VIII	1,621	332	35,466.06	5.00	3	2	
FaTLP62	6,391,832	6,394,818	Chr7-3	V	2,031	661	72,913.56	5.38	8	7	
FaTLP63	7,013,691	7,015,141	Chr7-3	II	1,156	362	39,230.87	8.62	4	3	
FaTLP64	6,395,823	6,402,472	Chr7-3	V	2,787	844	92,124.41	5.85	9	8	
FaTLP65	4,87,772	4,89,436	Chr7-4	VIII	1,474	332	35,497.10	5.00	3	2	
FaTLP66	5,545,850	5,547,249	Chr7-4	II	1,089	362	39,081.68	8.42	3	2	
FaTLP67	6,225,650	6,227,172	Chr7-4	II	996	331	35,782.12	7.45	3	2	
FaTLP68	4,792,251	4,809,304	Chr7-4	V	4,370	1,255	1,38,401.93	6.19	18	17	
FaTLP69	4,018,164	4,020,305	Chr2-3	VI	1,134	377	39,155.99	4.62	3	2	
FaTLP70	24,864,686	24,865,432	Chr3-2	III	747	248	26,635.50	8.58	1	0	
FaTLP71	22,604,600	22,605,641	Chr4-2	VIII	726	241	24,694.60	4.62	2	1	
FaTLP72	29,107,013	29,108,091	Chr4-3	VIII	726	241	24,674.58	4.48	2	1	
FaTLP73	1,058,206	1,058,952	Chr5-1	V	747	248	26,881.21	5.55	1	0	
FaTLP74	24,100,394	24,101,427	Chr5-3	VIII	921	247	26,326.74	4.75	2	1	
FaTLP75	16,618,205	16,619,717	Chr6-2	VIII	1,127	297	32,080.80	4.28	3	2	
FaTLP76	24,122,820	24,126,239	Chr7-2	V	2,392	230	24,400.62	6.70	2	1	
Note:

Detailed information on the FaTLP genes, including their names, chromosome position, CDS lengths, molecular weights, and pI values.

Phylogenetic relationships of TLPs in major plant species

To investigate the evolutionary relationship of the TLP gene families, 10 representative Arabidopsis thaliana and Oryza sativa TLPs sequences were used as reference sequences together with F. × ananassa and F. vesca TLPs in the phylogenetic analysis. Similar to the phylogenic results of Arabidopsis and Oryza TLPs (Shatters et al., 2006), the 76 FaTLPs were also classified into 10 phylogenetic groups, Groups I to X (Fig. 1A). A maximum number (16) of FaTLPs were clustered in Group VIII, followed by Group V (15), and these two groups accounted for almost half of the FaTLPs. Groups III and IX contained the least amount of TLP family genes with two genes each. Similarly, most of the F. vesca TLPs were concentrated in Groups VIII and V, and their distribution was more uneven than the FaTLPs.

Figure 1 Phylogenetic analysis and chromosome distribution analyses of the TLP gene family.

(A) A phylogenetic tree was constructed for F. × ananassa, F. vesca, Arabidopsis thaliana, and Oryza sativa using MEGA 6.0 with bootstrap values of 1,000. All of the tested TLP genes were divided into 10 groups (Groups I to X), represented by different colors. (B) The distribution of FaTLP on 24 different chromosomes.

Chromosomal distribution of TLP family genes in F. × ananassa

To explore the distribution of TLP family genes on the chromosomes of F. × ananassa, we performed a chromosomal localization analysis. The results showed that the FaTLP members were distributed on 24 (42.9% of 56) F. × ananassa chromosomes (Fig. 1B). The number of FaTLP genes for each chromosome varied significantly. The largest number of genes (seven) was detected on Chr6-1; followed by six genes on Chr2-1, Chr2-3, and Chr7-2; five genes on Chr7-1 and Chr7-4; and the fewest were found on Chr3-2, Chr3-4, Chr4-2, Chr4-3, and Chr5-2 (one per chromosome). Similar chromosomal distribution has been shown in other species, Arabidopsis TLP genes were distributed on seven (29.2% of 24) chromosomes, and there were 25 (56.8% of 44) in rice and 28 (57.1% of 49) in poplar TLP genes (Cao et al., 2016). Additionally, the same paraphyletic group of genes were not distributed on a certain chromosome; that is, all seven genes in Group VII were distributed on seven chromosomes (Chr1-1, Chr1-2, Chr1-3, Chr1-4, Chr2-1, Chr2-2, and Chr2-3), and all nine genes in Group II were distributed on seven chromosomes (Chr6-1, Chr6-3, Chr6-4, Chr7-1, Chr7-2, Chr7-3, and Chr7-4). Segmental duplications might be a major factor that contributed to these distributional properties (Cannon et al., 2004). However, there are a few exceptions. For example, five genes from Group VIII were distributed on the same chromosome (Chr6-1), suggesting that they may come from the tandem duplications (Cao et al., 2016; Faillace et al., 2019). Our results indicated that the expansion and diversification of the TLP gene family in F. × ananassa might be caused by tandem and segmental duplication events. Moreover, the genes were not evenly distributed on a certain chromosome and more genes were distributed at both ends of the chromosome. For example, Chr1-1, Chr2-2, Chr2-3, Chr3-2, Chr4-2, Chr4-3, Chr5-4, Chr6-3, Chr7-1, and Chr7-2 were closer to the ends, while Chr5-2 and Chr6-4 were near the centromere. This character may be due to the distribution of more repeat sequences in the centromere (Wang et al., 2020).

Conserved motifs of TLP genes

The diversity of the motif compositions of TLP genes in F. × ananassa was assessed using MEME software, and a total of 15 conserved motifs were obtained. The distribution of these 15 motifs in the TLPs are shown in Fig. 2. FaTLP10 and FaTLP68 contained all 15 conserved motifs. In addition, four genes (FaTLP55, FaTLP56, FaTLP59, and FaTLP62) contained 14 motifs but did not include motif11. Motif1, 2, 3, 4, 5, 6, 7, 10, 11, and 15 were found in most of the FaTLP genes. Furthermore, conserved motif6 was widely distributed across all 76 FaTLP genes. Motif2 and motif7 were absent only in one gene (FaTLP5), and motif3 was not found in two genes (FaTLP9 and FaTLP33). These 10 conserved motifs were also common across all groups (Group I to X). Moreover, some members of Group V (FaTLP10, FaTLP11, FaTLP55, FaTLP56, FaTLP58, FaTLP59, FaTLP62, FaTLP64, and FaTLP68) shared several unique motifs, namely motif8, 9, 12, 13, and 14. These results suggested that the TLP genes in each group shared several unique motifs and may have certain functional similarities. Moreover, these motifs were relatively conserved, which is why they may be used as markers for the identification of TLP genes and important functional components of the TLP gene family.

Figure 2 The conserved motif analyses of the TLP proteins of F. × ananassa.

(A) The motif compositions were predicted using MEME software, and the 15 conserved motifs are represented by different colors. (B) The sequence logos of all 15 motifs of the FaTLP.

Transcriptome changes in different resistant strawberry cultivars in response to C. gloeosporioides infection

To better understand the transcriptome profile of different resistant strawberry cultivars in response to C. gloeosporioides, RNA-seq analysis was used on 12 samples (‘Kaorino’-infected, ‘Kaorino’-uninfected, ‘Benihoppe’-infected, and ‘Benihoppe’-uninfected) with three replicates per treatment. Approximately 576 million raw reads were obtained, and the clean reads were mapped to the F. × ananassa genome (Table S1). Based on p-adjust < 0.05, and log2FC ≥ 1, a total of 10,462 and 13,682 DEGs were detected in the ‘Kaorino’-infected/uninfected (resistant (R) group) and ‘Benihoppe’-infected/uninfected (susceptible (S) group) leaves, respectively. Additionally, 5,490 (40.12%) genes were downregulated in the S-group compared to 4,765 (45.55%) in the R-group (log2FC ≤ −1) (Fig. S1, Table S2). More genes were upregulated in the S-group (8,192, 59.87%) than in the R-group (5,697, 54.45%) (log2FC ≥ 1) (Fig. S1, Table S3). Based on sequence homology, the DEGs were classified into 48 functional groups belonging to three main GO ontologies: cellular components, molecular functions, and biological processes. Among these DEGs, 2,993 (28.61%) genes in R-group and 3,564 (26.05%) genes in S-group were involved in the GO categories “response to stimulus” (Fig. S2), which contained three subgroups associated with fungal resistance: GO:0050832, GO:0009817, and GO:0009620 (Fig. S3). Six DEGs annotated as TLPs were categorized into these functional subgroups, indicating that these DEGs varied greatly in response to C. gloeosporioides (Table S4). According to our identification of the TLP gene family in octoploid strawberry, the six differentially expressed TLP genes were identified as FaTLP40, FaTLP41, FaTLP43, FaTLP62, FaTLP68, and FaTLP75. The RNA-seq results showed that at 24 hpi, FaTLP40 and FaTLP41 were upregulated and FaTLP75 was downregulated in the resistant group, while FaTLP62 was upregulated in the susceptible group, and FaTLP43 was downregulated and FaTLP68 was upregulated in both groups.

qRT-PCR analysis of TLP genes in response to C. gloeosporioides infection

To further understand the roles of TLP genes in strawberry, we investigated the expression profiles of FaTLP genes in F. × ananassa. The qRT-PCR results of these TLP genes in different resistant strawberry cultivars after infection with the fungal pathogen showed a wide range of expression responses (Fig. 3B, File S1). The induction expression of FaTLP68 was significant in the resistant cultivar (‘Kaorino’) at 12 hpi, although it decreased over time, whereas the upregulation of FaTLP68 was detected at 24 hpi in the susceptible cultivar (‘Benihoppe’), and its expression peak appeared later than that of the resistant cultivar. FaTLP40 and FaTLP41 were also highly induced in both strawberry cultivars but there were two expression peaks. This induction increased over time and reached its first peak at 24 hpi and the second peak at 4 d post-inoculation (dpi) in the resistant cultivar, whereas the susceptible cultivar had its first expression peaks of FaTLP40 and FaTLP41 at 48 hpi. The expression of FaTLP62 was considerably upregulated at 24 hpi only in the susceptible cultivar. These results showed a trend similar to the one observed using RNA-seq. Nevertheless, a gradual downregulation was observed for FaTLP43 and FaTLP75 in ‘Kaorino’ after pathogen infection, which then significantly increased at 12 hpi and 48 hpi, respectively, followed by a decrease until 7 dpi. This was not entirely consistent with the results of the transcriptome analysis, indicating that FaTLP43 and FaTLP75 were downregulated at 24 hpi. However, in the susceptible cultivar, the expression patterns of FaTLP43 and FaTLP75 were similar during the initial stage after inoculation but the maximum peak occurred at 4 dpi (1 day after disease symptoms were visible to the naked eye; Fig. 3A). Overall, the qRT-PCR analysis indicated a clear upregulation of the six abovementioned TLP genes following C. gloeosporioides infection in the octoploid strawberry.

Figure 3 Time course of different resistant strawberry leaves and the qRT-PCR results of TLP gene responses to C. gloeosporioides infection.

(A) The progression of symptoms between ‘Kaorino’ and ‘Benihoppe’ after C. gloeosporioides inoculation. Visible symptoms appeared earlier and progressed faster in ‘Benihoppe’. (B) The expression analysis of FaTLP (FaTLP40, FaTLP41, FaTLP43, FaTLP62, FaTLP68, and FaTLP75) in different resistant strawberry cultivars after infection. The x-axis represents the different times post-inoculation. The y-axis represents relative expression. Black and gray sections represent ‘Kaorino’ and ‘Benihoppe’, respectively. The results were normalized against the reference gene and shown as means of three replicates ± SDs.

Discussion

TLPs are a PR protein family that plays a key role in plant defense. In this study, 76 TLP gene members were identified in F. × ananassa, and their characteristics, phylogenetic relationships, motif organization, and chromosomal location were investigated. Using reference sequences from Arabidopsis and Oryza, F. × ananassa TLP genes were clustered into 10 paraphyletic groups, although the distribution of the TLP genes in each group was not uniform. Phylogenetic studies using Arabidopsis, Oryza (Shatters et al., 2006), Populus (Zhao & Su, 2010) and barley (Iqbal et al., 2020) species found that TLPs were also unevenly distributed across the groups, whereas the largest groups contained 11 (36.67%), 5 (23.81), 25 (45.45%), and 7 (36.84%) TLPs, respectively. Previous studies hypothesized that those clades containing the largest gene members would be more active against pathogens or have other exogenous stimuli and more rapid adaptive evolution (Zhao & Su, 2010). In this study, Groups V and VIII were the largest gene clades in our paraphyletic results. Our reference TLP sequence (AT4G11650) from Arabidopsis, annotated as an osmotin-like gene ATOSM34, was found in Group V. Three other PR TLP genes from Arabidopsis (At5g24620) and Oryza (Os07g23470 and Os08g40600) involved in defense response to pathogens (Ascencio-Ibáñez et al., 2008; Liu, Zamani & Ekramoddoullah, 2010) were also clustered in Group V and Group VIII, although their exact functions have not been verified. Osmotin is a TLP belonging to the PR5 family that was originally regarded as a salt-induced protein (Singh et al., 1989), and its antifungal mechanism of action has been studied in detail (Xu et al., 1994; Ibeas et al., 2001; Salzman et al., 2004). ATOSM34 has been reported as a key Arabidopsis gene involved in the defense response to biotic and abiotic stress (Capelli et al., 1997; Mukherjee et al., 2010; Vibhuti et al., 2016; Park & Kim, 2021). We used transcriptome data to identify the six genes encoding TLP proteins associated with plant responses to C. gloeosporioides infection, FaTLP40, FaTLP41, FaTLP43, FaTLP62, FaTLP68, and FaTLP75, which were also clustered in Group V (FaTLP40, FaTLP41, FaTLP62, and FaTLP68) and Group VIII (FaTLP43 and FaTLP75), respectively. These FaTLP genes clustered in the same clade as the reference genes, which have anti-fungal functions, indicating that the TLP genes belonging to the same phylogenetic group may have certain functional similarities.

Further qRT-PCR analysis verified that five of the TLP genes were up-regulated in resistant strawberry cultivars upon C. gloeosporioides infection (Fig. 3). Upregulation or overexpression of TLP genes often results in enhanced antifungal activity against several different pathogenic fungi (Datta et al., 1999; Fagoaga et al., 2001; Kalpana et al., 2006). For example, overexpression of barley TLP-1 in transgenic wheat lines improved pathogen resistance to Fusarium graminearum (Mackintosh et al., 2007). Similarly, increased TLP gene expression in transgenic tobacco plants enhanced resistance to Pythium aphanidermatum and Rhizoctonia solani (Rajam et al., 2007), while overexpression of the TLP gene VaTLP improved downy mildew resistance in Vitis vinifera (He et al., 2017). In turn, significant upregulation of FaPR5-1 and FaPR5-2 was observed in the salicylic acid-primed defense response of octoploid strawberry to Podosphaera aphanis (Feng et al., 2020). Ultimately, these results suggest that FaTLP40, FaTLP41, FaTLP43, FaTLP68, and FaTLP75 may have potential functions in plant resistance responses to C. gloeosporioides.

Our findings demonstrated an initiation of transcriptional responses in strawberry leaves during C. gloeosporioides invasion for some FaTLP genes. Additionally, the expression patterns of these genes revealed that the two different resistant strawberry cultivars differed in their temporal defense responses. FaTLP43 and FaTLP75 activated defense responses much faster in ‘Kaorino’ (<48 hpi) than in the susceptible cultivar ‘Benihoppe’ (4 dpi). Therefore, we hypothesized that during C. gloeosporioides inoculation in strawberry leaves, the spread of the fungus was suppressed because of a early defense response in the resistant cultivar while the pathogen spread out of control in the susceptible cultivar due to the delayed defense response. Similar results were also reported in the response of strawberries to Verticillium dahliae (Besbes, Habegger & Schwab, 2019) and Podosphaera aphanis (Feng et al., 2020) infection, which supports our hypothesis. The early activated defense mechanism of these TLP genes in resistant cultivar such as ‘Kaorino’ needs to be further studied. Our results provide a new strategy for developing anthracnose-resistant strawberry cultivars in the future by promoting early response TLP gene expression with antifungal activity in susceptible cultivars.

Conclusion

In this study, we performed genome-wide identification and characterization of TLPs in octoploid strawberry. A total of 76 TLP genes (FaTLP1–76) were identified using genome-wide screening. Comparative phylogenetic analysis divided the TLPs into 10 groups and the functions of TLPs in F. × ananassa were analyzed. Our qRT-PCR analysis indicated a clear upregulation of five TLP genes in resistant strawberry leaves infected with C. gloeosporioides. Furthermore, our results showed differences in TLP gene expression patterns between two different resistant strawberry cultivars. We concluded that the TLPs’ early activated defense to pathogenic fungi might be a reason why the resistant strawberry cultivar ‘Kaorino’ showed greater anthracnose resistance than the susceptible cultivar ‘Benihoppe’. This study lays the foundation for further exploration of the antifungal function of TLP genes in F. × ananassa, and provides new strategies for improving strawberry resistance to anthracnose through genetic engineering.

Supplemental Information

Supplemental Information 1 RNA-Seq reads and reads mapping.

Click here for additional data file.

Supplemental Information 2 Differentially expressed genes in Kaorino-infected/uninfected.

Based on p-adjust < 0.05, and |log2FC| > 1.

Click here for additional data file.

Supplemental Information 3 Differentially expressed genes in Benihoppe-infected/uninfected.

Based on p-adjust < 0.05, and |log2FC| > 1.

Click here for additional data file.

Supplemental Information 4 Differentially expressed genes of TLP.

Based on p-adjust < 0.05, and |log2FC| > 1.

Click here for additional data file.

Supplemental Information 5 Primer sequences derived from qRT-qPCR analysis.

The Actin gene was used as a reference gene. Each sample was repeated in triplicate.

Click here for additional data file.

Supplemental Information 6 The number of up-or downregulated DEGs 24 h post inoculation (hpi) in both groups.

The x-axis represents the different treatments. The y-axis indicates the number of DEGs. Black and gray sections represent upregulated and downregulated genes, respectively.

Click here for additional data file.

Supplemental Information 7 Histogram of the Gene Ontology (GO) classifications of ‘Kaorino’-infected/uninfected and ‘Benihoppe’-infected/uninfected.

The x-axis represents enriched GO processes, and different colors represent different GO processes. The y-axis indicates the total number of genes annotated to each GO process.

Click here for additional data file.

Supplemental Information 8 Histogram of DEGs associated with fungal resistance in three GO process subgroups.

The x-axis represents the number of DEGs. The y-axis indicates the three subgroups associated with fungal resistance. Black and gray sections represent ‘Kaorino’-infected/uninfected and ‘Benihoppe’-infected/uninfected, respectively.

Click here for additional data file.

Supplemental Information 9 Raw data for qRT-PCR.

Click here for additional data file.

We thank Dr. Li Fang and Dr. Yunye Xie of the Institute of Plant Protection and Microbiology, Zhejiang Academy of Agricultural Sciences for providing the pathogenic fungus C. gloeosporioides used in this study.

Additional Information and Declarations

Competing Interests

Author Contributions

Data Availability

The authors declare that they have no competing interests.

Yuchao Zhang conceived and designed the experiments, performed the experiments, analyzed the data, prepared figures and/or tables, authored or reviewed drafts of the paper, and approved the final draft.

Lixiang Miao performed the experiments, analyzed the data, prepared figures and/or tables, and approved the final draft.

Xiaofang Yang performed the experiments, prepared figures and/or tables, and approved the final draft.

Guihua Jiang conceived and designed the experiments, analyzed the data, authored or reviewed drafts of the paper, and approved the final draft.

The following information was supplied regarding data availability:

The raw data of the RNA-seq experiment is available at NCBI Sequence Read Archive: PRJNA754844.

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
