# Peer review of "Genome-wide characterization and expression of the TLP gene family associated with Colletotrichum gloeosporioides inoculation in Fragaria × ananassa"

_PeerJ, doi:10.7717/peerj.12979_

## Round 0.1 · original submission · Major Revisions

Dear Dr. Zhang,

Your manuscript had reviewed by three experts in your research areas. Based on the reviewers’ comments and my assessment, your manuscript requires substantial revisions. Some shortcomings in your manuscript are:

(i) lack of clear statement of justifications - why this research (e.g., TLP genes) was necessary;

(ii) clearly describe specific objectives with interconnected working hypotheses;

(iii) in Materials and Methods, explain the proper experimental design, and sampling strategies;

(iv) in the result section, precisely reorganize TLP genes in response to C. gloeosporioides resistance;

(v), Discussion section needs to be improved sustainably and must be concise and precise; (

vi) all figure qualities need to improve, and tables must be self-explanatory,

(vii) extensive English editing is necessary throughout the manuscript for clarity.


For further comments, please see the attached pdf file and reviewers' comments below.

·

Basic reporting

The introduction section needs to provide, generalized background of the topic and explanation for why TLPs are highly complex gene family. However, to make the manuscript more substantial, the author may need thorough polishing of English. I suggest major revision before accepting for publication. Some more comments are mentioned below
I) Rephrase line 124-125 shock
II) Figure 1A needs to be more legible
III) In Table 1 the information of chromosome needs to be incorporated
IV) Rephrase line 194-197
V) Rephrase line 206-210
VI) Line 239, FaTLP68 is predicted as calmodulin-binding receptor-like cytoplasmic kinase 2 isoform X2 and not as TLP. If it is not TLP then why the authors have selected these gene for further validation.
VII) Raw data of qRT-PCR not cited in the manuscript
VIII) Line 276, Shatters et al., 2006 is missing in reference
IX) Discussion section needs to be improved further, especially for phylogenetic and motif studies

Experimental design

I think the motivations for this study need to be made clearer. In particular, the connection between the phylogenetic and motif analysis, and the gene expression, could be clearer. One way to demonstrate this connection would be to relate which phylogenetic clade genes have higher infection due to C. gloeosporioides infection. Furthermore, I was also wondering why the author chose outdated neighbor joining method for the construction of phylogenetic tree, the author also offers no explanation of why he included FaTLP68 for expression study, even though it was not annotated TLP.

Validity of the findings

The authors reported TLP genes in Fragaria ×ananassa and their importance for defense against C. gloeosporioides which seems interesting Genome-wide identification of these TLP reported genes might be helpful for future research.

Reviewer 2 ·

Basic reporting

The manuscript submitted has an acceptable level of English however, there were some lagging space in the manuscript, therefore, suggested to go through the manuscript before final submission for some grammar and technical errors in the word file.

Line 228-234: In addition to the numbers of gene also mention their percentage as well in brackets.

Line 289-290: Not all six genes overexpressed in resistant cultivar, FaTLP62 upregulated in susceptible cultivar only. should be rephrased

Experimental design

1. The comparative phylogenetic analysis carried out to identify the functional relationship of strawberry TLPs with TLPs from A. thaliana and O. Sativa could be explained more.

2. The distribution of FaTLPs on the 24 chromosomes need more explanation as why in most cases the same group is scattered on the different chromosomes. Is this the same with A. Thaliana and O. Sativa?

Validity of the findings

Line 228: Why more genes were upregulated in susceptible genotype compared to resistant one? Can you explain it more with any reference?

Line 278: explain further with reference.

·

Basic reporting

It is nit clear why this specific study was needed? What specific knowledge gap existed?

Experimental design

Methods are not described with sufficient detail.

Validity of the findings

No comment.

Additional comments

Manuscript is well written and authors have demonstrated good methodology in addressing the research object. However, there are several issues that the authors need to address before this manuscript can be published. Here are my comments:
- Line 72 - 77 (Paragraph 4 sentence 1 to 3) - Here, just before the Objectives statement, there should be a concise Rationale statement (Why this specific study was needed?). What specific knowledge gap existed?
- Line 77 to 80 (paragraph 4 sentence 4 to 6) - This reads like “Materials and methods” when it should be a statement of objectives.
- Line 123 – 126 – preparation of C. gloeosporioides cultures, this is not correct as stated in this section, kindly revise this method. Revise this section clearly. It is not clear in what form was the C. gloeosporioides provided. The culture medium is supposed to be potato dextrose broth.
- Line 128 – What was the basis of using 60-day old plants as the age of plants for fungal inoculation?
- Line 130 - Control plants were inoculated with sterile water – spraying with sterile water cannot be referred to as inoculation. In addition, why did the authors use sterile water instead of spraying with non-pathogenic C. gloeosporioides as controls? The authors would have autoclaved the C. gloeosporioides suspensions and use as controls.
- Significant sections of the methods of the MS are profoundly unclear
- Lines 175 – 177 - What was the basis of selecting these plant species Arabidopsis thaliana, Oryza sativa, and Fragaria vesca to determine the evolutionary relationship of TLP gene families.
- Lines 272 – 277 - So, what new have you discovered then? You are basically confirming what others have already found. This corroboration with a previously study is not that important.
- Lines 280 – 286 – This is literature review and not discussion of your results. This section should be transferred to the introduction section.
- Lines 303 – 305 - What are the implications of all this in the control C. gloeosporioides? The information you are talking about is already present in Figures. Discussion of implications is more important.
- Lines 312 – 314 – You are not sure. This is speculation! We need to know the implications of the findings for the control C. gloeosporioides program.
- In the Discussion, you have been emphasizing more on your agreement with other studies than on the novelty of your own findings. How does your agreeing with others help your program to control C. gloeosporioides? You should highlight what new you have discovered.

- Lines 323 – 324 – The statement is not correct since the authors did not determine differences in TLP gene expression patterns among different resistant strawberry cultivars. It was between one susceptible and one resistant.
- The quality of figures are low and needs to be revised.

---

## Round 0.2 · accepted · Accept

Dear authors:

Your revised manuscript, which you submitted to PeerJ, has been reviewed by two experts in your research area. Both reviewers had recommended for publication in PeerJ.

Thank you for considering publishing your research in PeerJ.

Best regards,

Sincerely,

Tika Adhikari

·

Basic reporting

The manuscript contributes to the knowledge of C. gloeosporioides infection in F×ananassa. The introduction and discussion section is improved as compared to the earlier version. Appropriate information about the earlier study is presented well for the readers to follow the rationale and findings of the study.

Experimental design

The experimental design for this manuscript is robust and acceptable for publication

Validity of the findings

The results presented in this manuscript will help in understanding the role of TLP against C. gloeosporioides.

Reviewer 2 ·

Basic reporting

The authors have have made the necessary changes as advised in the previous version.

Experimental design

The experimental design has been added in more detail now and is satisfactory.

Validity of the findings

The comments and required references have been incorporated in the discussion part.

Additional comments

The authors have done a good job of revising the manuscripts as per the instructions.